# EEG-RAGNET: RETRIEVAL-AUGMENTED GRAPH STRUCTURE REFINEMENT FOR CLINICAL SEIZURE DIAGNOSIS

## ABSTRACT

Seizure diagnosis from EEG signals is a critical yet persistently challenging task, due to the complicated neural dynamics and the spurious connections in inter-channel modeling. While spatial-temporal graph neural networks (STGNNs) have advanced EEG brain network representation learning, the resulting graph structures suffer from low clinical plausibility and limited interpretability due to their purely data-driven nature. To this end, we introduce EEG-RAGNet, a retrieval-augmented graph refinement framework that incorporates external medical knowledge to calibrate noisy EEG graphs. We first construct a large-scale, domain-specific knowledge base derived from authoritative clinical guidelines. Leveraging large language models, we extract structured biomedical entities and relations to form a textual knowledge graph (KG), which serves as external knowledge source of clinical priors. Our framework performs alignment-aware query construction by projecting STGNN-generated EEG node embeddings into the semantic space of KG. Semantic queries are then executed via FAISS-based similarity search over knowledge triplets to retrieve relation evidence. Each predicted edge is assigned a confidence score based on retrieved similarity, relation type, and source reliability, enabling us to prune medically implausible edges from the originally predicted graph. Extensive experiments on TUSZ and CHB-MIT demonstrate that EEG-RAGNet not only improves seizure detection accuracy but also enhances interpretability by grounding each prediction in clinically validated knowledge. This work provides the first unified framework that tightly couples brain dynamics with external medical expertise via retrieval-augmented reasoning, paving the way for knowledge-enhanced, explainable clinical diagnosis. The code is available at: https://anonymous.4open.science/r/EEG-RAGNet-63EE/.

## 1 INTRODUCTION

Epilepsy is one of the most prevalent neurological disorders, characterized by abnormal electrical discharges in the brain that can lead to recurrent and spontaneous seizures Weng et al. (2025); Tang et al. (2023). Among all diagnostic approaches, electroencephalography (EEG) remains the most widely used modality for capturing and analyzing brain activity related to epileptic seizures due to its non-invasive nature and high temporal resolution Ho & Armanfard (2023). However, the complex, nonlinear, and noisy nature of EEG signals presents significant challenges for accurate seizure detection and diagnosis Xiao et al. (2024); Cai et al. (2023). Importantly, seizure activity typically begins in a focal area and spreads through the brain's functional network to distant but connected regions Liu et al. (2022). The spatial spreading of seizure activities closely aligns with the brain's functional network structure Chen et al. (2023), suggesting that graph representation modeling can serve as a natural and physiologically meaningful approach for seizure analysis. In particular, spatial-temporal graph neural networks (STGNNs) have recently emerged as a promising class of methods for modeling dynamic brain networks from EEG data, as they can effectively capture both temporal dynamics and spatial dependencies across electrodes.

Although they show promising performance and the ability to model non-Euclidean spatial connectivity across brain networks, existing STGNN-based EEG representation modeling faces several critical limitations. First, the learned brain network graphs often contain noisy or clinically implau-

sible connections, resulting in redundant structures that may mislead downstream predictions Xie et al. (2023). Second, current approaches heavily rely on data-driven learning, lacking explicit integration of domain medical knowledge such as seizure patterns or diagnostic criteria, which severely limits the interpretability and clinical trustworthiness Díaz-Montiel et al. (2024). Third, some inter-channel connections exhibit low confidence or ambiguous predictive value. These 'borderline' connections are difficult to validate based solely on signal patterns Xiang et al. (2025). Simply stacking more complicated neural network layers cannot effectively address these issues. Instead, there is a pressing need for knowledge-informed modeling that can assess and refine graph structures based on clinically grounded evidence.

To address these limitations, we seek to incorporate external medical knowledge to assess and refine graph structures. However, manually crafting such knowledge-driven rules is infeasible. This motivates leveraging large language models (LLMs), which have shown remarkable capacity in knowledge retrieval and reasoning. While LLMs such as GPT, LLaMA, and PaLM demonstrate impressive general reasoning and language capabilities, they often struggle in specialized domains due to their lack of task-specific alignment and domain grounding. In clinical applications like EEG-based seizure detection, these limitations become critical, as the models may produce confident yet clinically irrelevant or incorrect outputs. Despite their strength in modeling complex temporal dependencies and contextual patterns Yu et al. (2024); Naveed et al. (2025); Jin et al. (2023), LLMs are typically pre-trained on general web-scale corpora and lack integration with expert-curated medical knowledge Singhal et al. (2025). This disconnect hinders their ability to interpret EEG signals in accordance with clinical guidelines. As a result, LLMs are prone to hallucinations or omission of essential biomarkers, which compromises the reliability and interpretability of their predictions Kim et al. (2025). Therefore, simply prompting an LLM with raw EEG features, without grounding in medical knowledge, remains insufficient for accurate and trustworthy seizure diagnosis.

To address the key limitations in dynamic brain graph modeling with STGNNs, namely the redundant or clinically implausible connections, and the absence of domain knowledge supervision, we propose EEG-REGNet, which leverages medical knowledge retrieval to refine graph-based EEG representations. Specifically, We first build a structured knowledge graph (KG) from certified epilepsy guidelines and literature, using LLMs to extract clinical triplets. Given raw EEG graphs learned by STGNNs at each time step, we project EEG channel embeddings generated by STGNNs into the KG semantic space via a proposed Semantic Alignment Query module. For each inter-channel edge, we retrieve the top-k most relevant clinical relations using FAISS approximate search Douze et al. (2024). Each edge is then assigned a knowledge-guided confidence score based on triplet existence, semantic similarity, and source reliability. Edges with insufficient evidence are pruned, yielding a refined and clinically aligned brain graph. The RAG-based pruning strategy enhances STGNNs' prediction robustness and physiological plausibility. EEG-RAGNet is model-agnostic and seamlessly integrates with various GNN backbones and LLM variants for generalized EEG graph learning. The main contributions of this work are summarized as follows:

- **Construction of a domain-specific knowledge base.** We build a large-volume, domain-specific knowledge base dedicated to EEG-based seizure diagnosis, curated from certified clinical guidelines and recognized literature, organized in JSON format.
- **Semantic Align Query for Entity Matching and Relation Retrieval.** A semantic alignment query mechanism is proposed, which projects EEG channel embeddings into the KG space for entity matching, and perform relation retrieval to assess edge plausibility using knowledge triplets.
- **A generalized RAG framework for clinical spatial-temporal modeling.** We develop a RAG-guided graph calibration strategy that integrates clinical priors into STGNN-predicted brain graphs, enabling the pruning of clinically implausible connections and enhancing interpretability without sacrificing predictive performance.

## 2 PRELIMINARIES

**Graph-Based EEG Representation:** EEG signals naturally exhibit non-Euclidean, graph-like characteristics due to the spatial and functional relationships among EEG channels (i.e., electrodes) Ho & Armanfard (2023). Thus, EEG signals can be modeled as a sequence of dynamic graphs. Specifically, at each time step $t$, the brain activity is represented as graph $\mathcal{G}_t = (\mathcal{V}, \mathcal{A}_t, \mathbf{X}_t)$, where $\mathcal{V} = \{\mathbf{v}_1, \mathbf{v}_2, ..., \mathbf{v}_N\}$ is the set of $N$ EEG channels as graph nodes. $\mathcal{A}_t \subseteq \mathcal{V} \times \mathcal{V}$ represents

the edge set at time $t$, capturing the dynamic relationships. Each node $\mathbf{v}_i$ is associated with a feature matrix $\mathbf{x}_t^i$, representing EEG signal characteristics of the channel at the current time window. $\mathbf{X}_t \in \mathbb{R}^{N \times T}$ denotes all node (channel) features at time step $t$.

**Graph Retrieval-Augmented Generation (GraphRAG).** GraphRAG extends classical RAG paradigm by structuring the external knowledge base in the form of knowledge graph rather than a flat collection of text passages Xiao et al. (2025). In this setting, nodes represent entities or concepts and edges encode their relations, forming knowledge graphs or other structured databases. A retriever then leverages graph traversal strategies (e.g., path search, neighborhood expansion, or graph neural encoders) to identify relevant subgraphs that capture both explicit and implicit relations Gong et al. (2025). The generator will integrate this graph-grounded context to produce answers or rationales that are more coherent, explainable, and logically consistent Gong et al. (2025).

**Problem 1** (RAG-enhanced Spatial-Temporal Graph Learning for EEG Seizure Diagnosis). *Given a sequence of dynamic EEG brain graphs $\{\mathcal{G}_t^{raw} = (\mathcal{V}, \mathcal{A}_t, \mathbf{X}_t)\}_{t=1}^T$, where $\mathcal{V}$ is the set of EEG channels, $\mathbf{X}_t$ denotes the node features, and $\mathcal{A}_t$ denotes the initial edge structure (predicted by a base STGNN) at time step $t$, and a structured external medical knowledge base $\mathcal{K}$ containing biomedical entities and relations, our goal is to learn a function:*

$$\mathcal{F} : (\{\mathcal{G}_t^{raw}\}_{t=1}^T, \mathcal{K}) \rightarrow \left(\{\mathcal{G}_t^{refined}\}_{t=1}^T, \{y_t\}_{t=1}^T\right)$$

*where $\mathcal{F}$ is formulated as a Retrieval-Augmented Generation (RAG) framework, which jointly refines the raw graph structures $\mathcal{G}_t^{raw}$ and predicts the seizure labels $y_t$ at each time step by incorporating knowledge-aware retrieval signals.*

## 3 METHODOLOGY

In this section, we introduce EEG-RAGNet, a knowledge-augmented graph structure refinement framework designed to improve the structural reliability of dynamic EEG brain network graphs. Our method aims to enhance graph learning-based structural predictions derived from data-driven STGNN models by integrating domain knowledge extracted from authoritative medical database in the form of textual knowledge graphs.

Figure 1 is the detailed architecture of our proposed EEG-RAGNet framework. The pipeline operates in two stages. First, given the input EEG signals, a base STGNN first generates the basic spatial-temporal EEG brain network graphs $\mathcal{G}_t^{raw}$. Meanwhile, each node (EEG channel) embedding $c_t^i$ is projected into the knowledge graph (KG) embedding space for semantic alignment. Retrieved medical triplets from external knowledge sources are then used to assess the plausibility of each edge via three scores: (1) existence match, (2) semantic proximity, and (3) source reliability. Based on the graded confidence, EEG-RAGNet prunes implausible edges, yielding refined graphs $\mathcal{G}_t^{refined}$ with improved clinical plausibility.

### 3.1 STGNN-BASED BASIC EEG GRAPH STRUCTURE PREDICTION

In the first stage of EEG-RAGNet, we aim to construct an initial sequence of spatial-temporal brain connectivity graphs that reflect the dynamic functional interactions across different brain regions during EEG recording. Let the EEG signal at time step $t$ be represented as a multichannel time series input $\mathbf{X}_t \in \mathbb{R}^{N \times T}$, where $N$ is the number of EEG electrodes and $T$ is the length of time window. We start by obtaining graph $\mathcal{G}_t^{raw} = (\mathcal{V}, \mathcal{A}_t, \mathbf{X}_t)$, where $\mathcal{V}$ is the node set corresponding to EEG channels, and $\mathcal{A}_t$ is the edge set indicating functional or statistical connectivity.

To obtain these initial graph structures, we leverage spatial-temporal graph neural networks (STGNNs) to capture the temporal evolution and spatial dependencies. These models are trained to learn edge weights based on signal similarity, temporal activation patterns, or predefined correlation metrics (e.g., coherence, mutual information, or phase-locking value).

The output of this step is a series of graphs $\{\mathcal{G}_{t_1}^{raw}, \mathcal{G}_{t_2}^{raw}, \ldots, \mathcal{G}_T^{raw}\}$, each representing the brain network at a specific time slot. While these graphs provide a learning-based representation of connectivity patterns, they suffer from noise, overfitting, or lack of physiological interpretability. Therefore, a subsequent refinement step is required to incorporate domain knowledge and ensure that the learned structures align with known neuro-physiological and clinical insights.

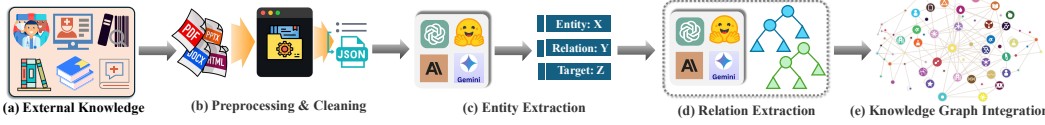

Figure 1: Illustration of EEG-RAGNet: a retrieval-augmented framework for refining EEG brain graphs using external medical knowledge.

## 3.2 Knowledge Base and Knowledge Graph Construction

**Knowledge Base Construction for EEG-based Epilepsy Diagnosis.** To establish a structured and reliable knowledge base for epilepsy diagnosis from EEG, we systematically curated and integrated information from a set of widely recognized and authoritative clinical guidelines. These guidelines represent international consensus and evidence-based recommendations, ensuring that the resulting knowledge base reflects both scientific rigor and clinical practicality. The primary knowledge sources come from the official website of: (1) International League Against Epilepsy (ILAE) Fisher (2017), (2) American Epilepsy Society (AES) Krumholz et al. (2015), (3) United Kingdom National Institute for Health and Care Excellence (NICE) Jones et al. (2023), (4) Scottish Intercollegiate Guidelines Network (SIGN) Soiza & Myint (2019), (5) Japanese Society of Neurology Chen et al. (2021). These guidelines represent the most authoritative, widely adopted international consensus and evidence-based practices, ensuring both scientific rigor and clinical validity. All documents are serialized into a unified corpus, where each entry corresponds to a page-level medical textual content. This provides a structured corpus of domain knowledge, forming the basis of RAG system.

(a) External Knowledge (b) Preprocessing & Cleaning (c) Entity Extraction (d) Relation Extraction (e) Knowledge Graph Integration

Figure 2: The pipeline of knowledge base construction, from medical sources to knowledge graph.

Building upon the knowledge base, we then employ GPT-4o model (via OpenAI API) to perform biomedical Named Entity Recognition (NER) over the entire guideline corpus. For each entry in the knowledge base, we crafted a structured prompt that instructed the model to extract entities across 12 semantic categories: (a) Diseases and Syndromes; (b) Symptoms and Clinical Manifestations; (c) Diagnostic Methods and Tests; (d) Medications; (e) Interventions and Procedures; (f) Anatomical Structures; (g) Risk Factors and Causes; (h) Biomarkers; (i) Biological Processes; (j) Clinical Measurements; (k) Medical Findings; (l) Actions and Treatments.

Each prompt asked the model to return its output in a strict JSON format, where keys are categories and values are arrays of detected entity mentions. An example prompt is given in Appendix C due to page limitations. This prompt-driven approach enabled scalable and consistent entity extraction across heterogeneous document formats and language styles, yielding a structured knowledge base of medically grounded concepts.

**Relation Extraction.** Following entity extraction, we perform relation extraction (RE) by instructing GPT-4o to identify explicit or implicit relationships between pairs of entities within the same text

segment. An example of our carefully designed relation extraction prompt is shown in Appendix C due to page limitations.

These extracted triplets are further validated for syntactic consistency and normalized to lowercase to reduce surface form variance. The final knowledge graph was then composed as a set of canonical (head, relation, tail) tuples, forming the basis for semantic matching and confidence estimation in subsequent modules.

## 3.3 SEMALIGNQUERY: SEMANTIC ALIGN QUERY CONSTRUCTION

To facilitate semantically meaningful retrieval from KG, we design a query construction mechanism, *SemAlignQuery*, which aligns local EEG graph substructures with the embedding space of medical knowledge. Instead of using a single pair of node embeddings as a query unit, we adopt a $k$-hop neighborhood-based approach to construct semantically enriched subgraph queries.

Given the EEG brain network graph $\mathcal{G}_t = (\mathcal{V}, \mathcal{A}_t, \mathcal{X}_t)$ at time step $t$, the STGNN encoder maps the input node feature $\mathbf{X}_t = \{x_t^1, x_t^2, ..., x_t^N\}$ into a set of latent embeddings $\mathbf{C}_t = \{c_t^1, c_t^2, ..., c_t^N\}$, where $c_t^i = \text{Enc}(x_t^i)$ denotes the embedding of EEG channel $i$. For each node $v_i \in \mathcal{V}$, we extract its $k$-hop local subgraph $G_t^{(i)} = \mathcal{G}_t[\mathcal{N}_k(v_i)]$, capturing context-aware topology within its neighborhood.

To represent this subgraph in the KG semantic space, we first obtain the embedding of each node $v_j \in \mathcal{N}_k(v_i)$ using a shared projection head $\phi_{\text{proj}} : \mathbb{R}^{d_{\text{STGNN}}} \to \mathbb{R}^{d_{\text{KG}}}$ as $\tilde{c}_t^j = \phi_{\text{proj}}(c_t^j)$. Although $c_i^t$ is already an embedding learned by the STGNN encoder, $\phi_{\text{proj}}$ is specifically designed to map it into the KG semantic space. This projection ensures cross-modal alignment between EEG feature embeddings and biomedical entity representations, which reside in heterogeneous vector spaces. The query vector $\mathbf{q}_i$ for $G_t^{(i)}$ is then constructed by aggregating the projected embeddings of all nodes in the subgraph, using a simple average:

$$\mathbf{q}_i = \frac{1}{|\mathcal{N}_k(v_i)|} \sum_{v_j \in \mathcal{N}_k(v_i)} \tilde{c}_t^j \tag{1}$$

The aggregated vector $\mathbf{q}_i$ is the semantic representation of the local EEG subgraph, and will thus be employed as the query vector for retrieval.

## 3.4 KNOWLEDGE-BASED RELATION RETRIEVAL

Once query vector $\mathbf{q}_i$ has been constructed from EEG channel embeddings, we proceed to retrieve semantically relevant medical relations from the knowledge base to refine the STGNN-predicted graph structure. We perform approximate nearest-neighbor retrieval in the KG to retrieve the top-$M$ most semantically similar knowledge triplets $\{\tau_m\}_{m=1}^M$, where each $\tau_m = (h_m, r_m, t_m)$ is a medical relation involving entities and clinical semantics.

To enable efficient retrieval, all triplet embeddings in the KG are pre-computed using BioBERT Lee et al. (2020) and indexed with FAISS Douze et al. (2024). FAISS (Facebook AI Similarity Search) is a high-performance library for large-scale vector similarity search and clustering, enabling fast nearest-neighbor retrieval even over millions of embeddings. The similarity between $\mathbf{q}_i$ and $\tau_m$ is computed via cosine similarity between $\mathbf{q}_i$ and the concatenated triplet representation $[\mathbf{e}_{h_m} \| \mathbf{e}_{t_m}]$:

$$\text{sim}(\mathbf{q}_i, \tau_m) = \frac{\mathbf{q}_i \cdot [\mathbf{e}_{h_m} \| \mathbf{e}_{t_m}]}{\|\mathbf{q}_i\| \cdot \|[\mathbf{e}_{h_m} \| \mathbf{e}_{t_m}]\|} \tag{2}$$

These retrieved triplets act as external knowledge cues for evaluating the clinical plausibility of connections within $G_t^{(i)}$, which are leveraged in the downstream graph structure calibration step. Each local subgraph $G_t^{(i)}$ is associated with its own set of top-M retrieved triplets $\tau_m$, which serve as shared evidential sources for all edges within that subgraph. The evidential attributes derived from these triplets are subsequently propagated to each $e_{ij}^{(t)} \in G_t^{(i)}$ to guide knowledge-based scoring.

**Relation Evidence Collection.** For each raw edge $e_{ij}^{(t)}$ in $\mathcal{G}_t^{\text{raw}}$, we collect relation evidence from the knowledge triplets $\{\tau_m\}_{m=1}^M$ retrieved for its corresponding $k$-hop subgraph $G_t^{(i)}$: (1) Relation Type:

the semantic type $r_m$ such as activates, inhibits, or associated_with; (2) ***Path Length***: the number of hops connecting $h_m$ and $t_m$ in the KG (e.g., 1-hop, 2-hop); (3) ***Evidence Strength***: confidence score or source priority (e.g., clinical guideline > peer-reviewed study > case report); (4) ***Similarity Score***: cosine similarity between the subgraph query vector $\mathbf{q}_i$ and the triplet embedding $[\mathbf{e}_{h_m} \| \mathbf{e}_{t_m}]$.

These evidential attributes are assigned to all edges within $G_t^{(i)}$. In the next stage, we incorporate these features to compute a confidence score for each edge, guiding the final graph structure.

## 3.5 RAG-Guided EEG Graph Structure Calibration

To refine the raw EEG-derived brain graphs, we assign each edge $e_{ij}^{(t)} \in \mathcal{G}_t^{\text{raw}}$ a knowledge-guided confidence score $\omega_{kg}(e_{ij}^{(t)}) \in [0, 1]$, which reflects how well the connection is supported by relevant medical knowledge. Specifically, for each edge $e_{ij}^{(t)}$ within a $k$-hop subgraph $G_t^{(i)}$, we retrieve a set of candidate medical knowledge triplets $\{\tau_m = (h_m, r_m, t_m)\}$ from the knowledge retriever. Each triplet serves as possible evidence explaining or rejecting the existence of $e_{ij}^{(t)}$. We compute the final confidence score $\omega_{kg}(e_{ij}^{(t)})$ by incorporating three critical factors: (i) *semantic similarity*, (ii) *source reliability*, and (iii) *relation-level alignment*.

$$\omega_{kg}(e_{ij}^{(t)}) = \text{sim}(\mathbf{q}_i, [\mathbf{e}_{h_m} \| \mathbf{e}_{t_m}]) \cdot r_m \cdot \mathbf{1}_{\text{match}}(e_{ij}^{(t)}, \tau_m) \tag{3}$$

Here, $\text{sim}(\cdot) \in [0, 1]$ denotes the cosine similarity between the subgraph query embedding $\mathbf{q}_i$ and the concatenated head/tail embeddings of the triplet. $r_m \in [0, 1]$ captures the credibility of the triplet source (e.g., clinical guidelines > research papers > case reports). The term $\mathbf{1}_{\text{match}}(e_{ij}^{(t)}, \tau_m)$ is an indicator that equals 1 if the entities aligned with $v_i$ and $v_j$ semantically correspond to $(h_m, t_m)$ in the triplet $\tau_m$, determined by entity-type matching and cosine-similarity thresholding; otherwise 0. When multiple triplets (top-$M$, $M > 1$) are retrieved, we compute the knowledge-guided confidence score $\omega_{kg}^m(e_{ij}^{(t)})$ for each triplet $\tau_m$ independently according to Eq. . The individual scores are then aggregated by averaging across all retrieved triplets to form the final edge confidence:

$$\omega_{kg}(e_{ij}^{(t)}) = \frac{1}{M} \sum_{m=1}^{M} \omega_{kg}^m(e_{ij}^{(t)}) \tag{4}$$

Edges with scores below a threshold $\rho$ are pruned from the original graph $\mathcal{G}_t^{\text{raw}}$, producing a refined graph $\mathcal{G}_t^{\text{refined}}$. This knowledge-guided pruning strategy enhances both the robustness and interpretability of EEG-based seizure analysis by enforcing alignment with trusted medical priors.

## 4 Experiments

We conduct extensive experiments to evaluate the effectiveness of our proposed EEG-RAGNet from multiple perspectives. The experiments are designed to answer the following Research Questions (RQ): (1) **RQ1:** Does EEG-RAGNet improve seizure detection performance over standard STGNN baselines? **RQ2:** Can RAG-based knowledge integration approach improve the quality and explainability of the learned EEG graph structure? **RQ3:** What is the individual contribution of each key component in EEG-RAGNet? **RQ4:** How does the choice of LLM for KG construction affect downstream seizure diagnosis performance?

### 4.1 Experimental Settings

**Dataset.** We evaluated our model on two public EEG seizure datasets: the Temple University Hospital EEG Seizure Corpus (TUSZ) v1.5.2 Shah et al. (2018) and the CHB-MIT Scalp EEG Database Shoeb (2009). TUSZ contains 5,612 EEG recordings with 3,050 annotated seizures from multiple subjects using 19 channels, making it one of the largest clinical EEG corpora. The CHB-MIT dataset includes 844 hours of 22-channel scalp EEG recordings from 23 pediatric patients (163 seizure events) sampled at 256 Hz. For both datasets, we followed their official or standard patient-wise splits, using seizure onset and offset annotations for evaluation. Due to page limitation, we provide a more detailed dataset descriptions in Appendix B.

Table 1: Performance comparison on the TUSZ and CHB-MIT datasets under different clip lengths. Bold values indicate the best performance within each setting.

| Dataset | Clip Len. | Methods | W EEG-RAG$_{Net}$ | | | W/O EEG-RAG$_{Net}$ | | |
|---|---|---|---|---|---|---|---|---|
| | | | F1(%) | Recall(%) | AUROC(%) | F1(%) | Recall(%) | AUROC(%) |
| TUSZ | 12-s | Dist-DCRNN | 71.3±0.9 | 73.5±2.3 | 87.1±1.8 | 70.3±1.1 | 71.6±2.4 | 85.0±1.7 |
| | | Corr-DCRNN | 72.6±1.1 | 75.6±1.4 | 86.3±0.9 | 70.7±1.0 | 73.4±2.2 | 85.4±2.1 |
| | | NeuroGNN | 64.7±1.3 | 71.0±1.9 | 86.5±2.1 | 62.6±1.4 | 70.4±1.3 | 83.7±1.9 |
| | | GraphS4mer | 69.0±0.9 | 72.1±1.3 | 87.2±1.5 | 66.5±0.8 | 71.5±1.8 | 83.1±1.2 |
| | | EvoBrain | **74.8±0.5** | **78.2±0.9** | **90.8±1.7** | **72.5±0.7** | **76.8±1.9** | **89.6±1.6** |
| | 60-s | Dist-DCRNN | 69.5±1.9 | 73.3±2.1 | 87.5±2.3 | 66.8±2.1 | 71.5±2.4 | 86.2±3.0 |
| | | Corr-DCRNN | 72.2±1.5 | 73.2±1.5 | 87.3±2.1 | 70.5±2.2 | 71.2±1.5 | 86.8±2.3 |
| | | NeuroGNN | 69.8±1.7 | 73.3±2.3 | 87.1±2.5 | 67.4±2.4 | 72.0±1.7 | 85.7±1.9 |
| | | GraphS4mer | 68.0±0.8 | 71.8±1.7 | 88.5±1.5 | 65.7±1.6 | 70.3±2.3 | 88.1±2.1 |
| | | EvoBrain | **75.3±0.7** | **78.8±0.9** | **91.6±1.1** | **72.8±0.9** | **76.4±1.8** | **90.3±1.6** |
| CHB-MIT | | Dist-DCRNN | 84.1±1.8 | 77.5±2.3 | 92.1±1.5 | 82.3±2.4 | 74.9±1.8 | 89.6±1.4 |
| | | Corr-DCRNN | 85.3±2.0 | 78.0±1.9 | 92.5±1.1 | 82.8±2.1 | 75.3±2.4 | 90.7±1.5 |
| | | NeuroGNN | 83.5±1.6 | 76.4±2.1 | 91.6±1.4 | 81.5±1.7 | 73.1±1.5 | 88.5±1.8 |
| | | GraphS4mer | 85.2±1.2 | 79.1±1.0 | 92.7±1.5 | 83.6±1.1 | 77.6±0.9 | 91.3±1.3 |
| | | EvoBrain | **89.3±0.9** | **83.2±1.2** | **94.5±1.4** | **87.5±0.8** | **80.7±1.3** | **91.6±1.1** |
| | | Dist-DCRNN | 85.3±1.5 | 78.1±1.9 | 92.5±2.0 | 82.6±1.9 | 75.6±2.1 | 89.2±1.6 |
| | | Corr-DCRNN | 85.8±1.3 | 79.0±1.7 | 92.3±1.6 | 82.2±1.8 | 76.2±1.4 | 89.8±1.7 |
| | | NeuroGNN | 84.6±1.4 | 77.8±2.1 | 91.8±2.2 | 81.5±1.6 | 75.4±0.8 | 88.7±1.3 |
| | | GraphS4mer | 86.3±1.3 | 80.6±1.1 | 93.5±1.6 | 83.2±1.4 | 78.1±2.2 | 91.6±1.4 |
| | | EvoBrain | **90.4±1.1** | **83.8±0.9** | **95.2±1.3** | **87.2±1.3** | **80.4±1.7** | **91.9±0.9** |

**Baseline Methods and Evaluation Metrics.** To comprehensively evaluate the performance of the proposed EEG-RAGNet, we compare it against a suite of STGNN models, covering both static and dynamic graph paradigms. (i) Static STGNN baselines: We evaluate Dist-DCRNN Tang et al. (2022), which builds a distance-based fixed connectivity graph and applies diffusion convolutional recurrent network. (ii) Dynamic STGNN baselines: We include Corr-DCRNN Tang et al. (2022) (Correlation-based DCRNN), GRAPHS4MER Tang et al. (2023), NeuroGNN Hajisafi et al. (2024), and EvoBrain Kotoge et al. (2025), all of which learn time-evolving brain network graphs to capture dynamic neural activities. For experimental evaluation, we adopt three widely used metrics: *Accuracy*, *F1 score*, and *AUROC*. Yang et al. (2023); Tang et al. (2022).

## 4.2 PERFORMANCE EVALUATION

Table 1 shows that integrating EEG-RAGNet consistently improves seizure detection performance across all STGNN baselines on both 12s and 60s EEG clips. Notably, dynamic graph learners such as Corr-DCRNN, NeuroGNN, and EvoBrain benefit more substantially compared to the static Dist-DCRNN, confirming that EEG-RAGNet is particularly effective in refining temporally-evolving but noisy connectivity patterns. For example, EvoBrain achieves the highest F1 and AUROC after knowledge integration (F1: 0.748→0.753, AUROC: 0.908→0.916), validating the utility of clinically grounded priors in enhancing graph quality. Although the absolute numerical gains appear modest (typically 1%-3%), such improvements are considered clinically and statistically meaningful in EEG seizure diagnosis, where data variability and inter-subject heterogeneity are inherently high. The consistent gain across all five representative STGNN architectures and both clip lengths further confirms the robustness and generality of our refinement mechanism.

Furthermore, we observe greater relative performance gains on shorter 12s clips, suggesting that when temporal context is limited, medical knowledge helps compensate by guiding more plausible spatial structures. These results demonstrate that EEG-RAGNet serves as a generalizable and modular enhancement to STGNN models, improving both the structural quality of brain graphs and the downstream diagnostic accuracy. Complementing this, Figure 3 (a)-(c) further demonstrate the impact of integrating EEG-RAGNet with various STGNN baselines. In (a), the combined models consistently outperform their original counterparts in seizure detection. (b) quantifies AUROC improvements brought by EEG-RAGNet across models. (c) reveals that EEG-RAGNet promotes sparser graph structures, suggesting enhanced focus on task-relevant connectivity patterns while suppressing noisy or redundant connections.

## 4.3 GRAPH STRUCTURE REFINEMENT AND INTERPRETABILITY ANALYSIS

**Decrease of Graph Redundancy.** Figure 3 (c) shows the average edge density of learned EEG graphs across baseline models, with and without EEG-RAGNet. We observe a consistent reduction in edge density after applying EEG-RAGNet, indicating its effectiveness in pruning noisy or

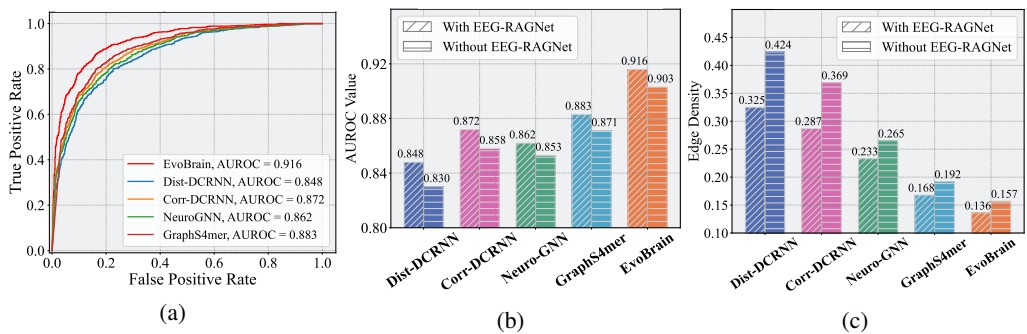

Figure 3: (a) ROC curves of baseline models integrated with EEG-RAGNet for seizure detection on 60s clips. (b) AUROC comparison of baseline STGNN models with and without EEG-RAGNet. (c) Edge density of learned brain graphs before and after EEG-RAGNet enhancement.

redundant connections. This refinement results in more compact and semantically meaningful graph structures, which improve interpretability and support better downstream seizure diagnosis.

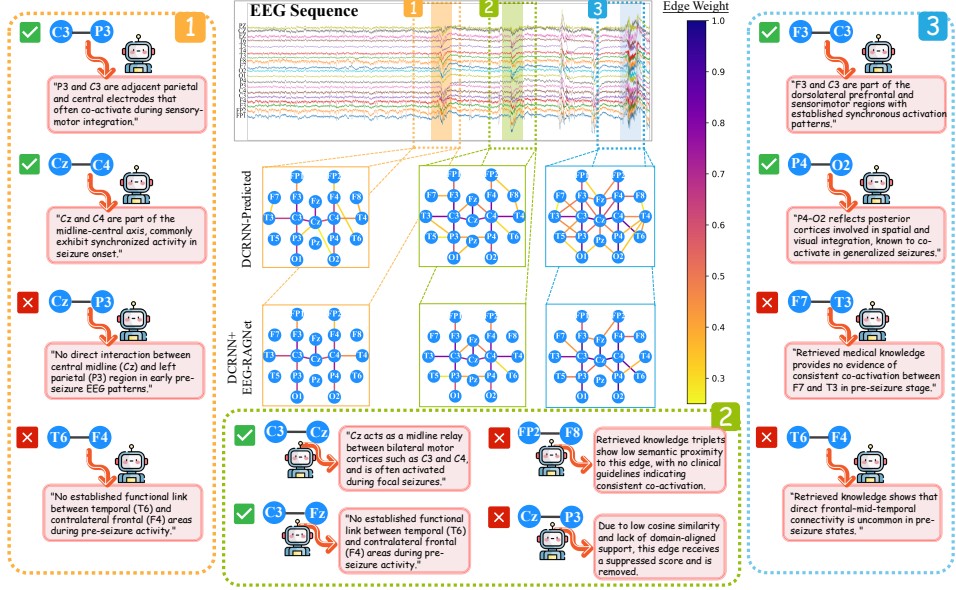

Figure 4: Visualization of EEG brain network graphs at three key pre-seizure moments, refined by our knowledge-enhanced EEG-RAGNet. The highlighted edges serve as case to demonstrate how domain knowledge supports or rejects specific functional connections between brain regions.

**Clinical Explainability.** Figure 4 illustrates how our EEG-RAGNet framework enhances the clinical plausibility of graph structures along an evolving pre-seizure EEG sequence. DCRNN is chosen as the representative STGNN backbone for our interpretability experiments. The first row of EEG brain network graphs are generated by DCRNN, which tends to over-connect nodes and form clinically implausible edges due to its purely data-driven nature. In contrast, the second row shows the refined graphs produced by our "DCRNN + EEG-RAGNet" pipeline, which prunes edges with weak medical support based on retrieved knowledge triplets.

In this figure, we conduct interpretability analysis on EEG graphs from three representative time points (noted as Case1,2,3), each capturing a clinically important phase within the pre-seizure progression. For example, in Case 1, edges like Cz-P3 and T6-F4 are incorrectly predicted by DCRNN but removed by EEG-RAGNet due to lack of known functional connectivity in early pre-seizure stages. In Case 2, spurious connections such as FP2-F8 are suppressed due to low semantic similarity and insufficient domain evidence. In Case 3, EEG-RAGNet preserves neurologically grounded

links (e.g., F3-C3, P4-O2) while eliminating unsupported ones (e.g., F-T3). This case-driven analysis underscores our model's ability to improve graph interpretability and reduce noise through medical knowledge integration.

## 4.4 ABLATION STUDIES

To better understand the contribution of the key components in EEG-RAGNet, we systematically remove or alterg specific modules and carefully design the following model variants: (I) **w/o SemAlignQuery:** It removes the semantic alignment module and directly uses raw EEG subgraphs as retrieval queries. (II) **w retrieval match only:** It simplifies the edge scoring process by using only the retrieval match score. (III) **w/o source reliability:** We eliminate the source reliability term from the final edge scoring, treating all retrieved evidence equally. *LLM Variants*: To assess the impact of different large language model on EEG-RAGNet, we replace the default GPT-4o with alternative LLMs during knowledge base construction, including (IV) **GPT-4o-mini**, (V) **GPT-3.5**, (VI) **Gemini 2.5 Flash**, (VII) **Qwen 2.5**, (VIII) **Llama3-8B**, and (IX) **Llama3-70B**.

In ablation study, we adopt EvoBrain Kotoge et al. (2025) as the backbone STGNN model and evaluate the impact of key components in EEG-RAGNet. As shown in Table 2, removing the SemAlignQuery module notably degrades performance, highlighting the importance of semantic alignment in forming precise and context-aware retrieval queries. The"retrieval match only" variant further confirms this by underperforming compared to the full model, suggesting that a multi-factor scoring strategy is crucial. Interestingly, excluding source reliability causes a slight drop in F1

Table 2: Ablation study on model variants.

| Model Variant | F1-score(%) | AUROC(%) |
|---|---|---|
| Full Model | **74.8**±**0.5** | **90.8**±**1.7** |
| w/o SemAlignQuery | 73.2±1.2 | 89.7±1.5 |
| w retrieval match only | 73.9±0.8 | 90.5±1.1 |
| w/o source reliability | 74.4±0.3 | 91.0±0.6 |
| w Llama3-8B | 70.6±1.9 | 85.8±1.6 |
| w Llama3-70B | 73.8±1.2 | 89.4±1.3 |
| w GPT-3.5 | 73.5±0.8 | 89.8±1.2 |
| w Gemini 2.5 Flash | 72.7±1.1 | 88.6±1.4 |
| w Qwen 2.5 | 71.9±0.9 | 87.3±1.5 |
| w GPT-4o-mini | 74.4±0.7 | 89.6±1.3 |

and AUROC, underscoring the importance of evaluating the trustworthiness of retrieved knowledge when refining edge predictions. Regarding LLM choice, using GPT-3.5 results in moderate performance decline, while smaller models like Llama3-8B and Llama3-70B lead to more significant drops, implying that both model quality and domain alignment matter. Overall, these results demonstrate the necessity of well-designed query, trust-aware scoring, and high-quality medical knowledge in maximizing the effectiveness of EEG-RAGNet.

## 5 CONCLUSIONS

In this work, we propose EEG-RAGNet, a novel retrieval-augmented graph learning framework designed to enhance dynamic brain graph construction for EEG-based seizure detection. Unlike prior STGNN approaches that rely purely on data-driven connectivity, EEG-RAGNet integrates external medical knowledge via a graph-based Retrieval-Augmented Generation (GraphRAG) pipeline, enabling more clinically grounded and interpretable brain network structures. Our approach constructs subgraph-level semantic queries based on k-hop neighborhoods, aligning learned EEG subgraphs with relevant neurophysiological knowledge. A multi-factor scoring strategy—combining match existence, semantic similarity, and knowledge source reliability—facilitates graph refinement by pruning anatomically implausible or noisy edges. Through extensive experiments on the TUSZ dataset, we demonstrate that EEG-RAGNet consistently improves seizure detection performance across a range of STGNN baselines. In particular, models with dynamic graph learning benefit more from knowledge-guided refinement. Experimental analysis further confirms the effectiveness of our framework in eliminating redundant connections while preserving discriminative structures. Moreover, the refined graphs not only improve classification accuracy but also enhance interpretability, laying the groundwork for trustworthy EEG-based decision support. EEG-RAGNet represents a step forward in bridging data-driven graph learning with domain knowledge in neuroscience, offering a generalizable paradigm for knowledge-enhanced neural decoding.

ETHICS STATEMENT

This work does not involve any personally identifiable information, or sensitive data. The Temple University Hospital EEG Seizure Corpus (TUSZ) dataset used in this work is a publicly available and anonymized dataset for research on epilepsy diagnosis. All experiments were conducted in accordance with established ethical guidelines, and no data was collected by the authors. Our research complies with the ICLR Code of Ethics, with a commitment to transparency, scientific integrity, and responsible dissemination. The proposed method is intended for clinical support, not as a substitute for medical expertise, and poses no foreseeable harm to individuals, communities, or environment.

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

# A  KNOWLEDGE SOURCES DESCRIPTION: RECOGNIZED EPILEPSY DIAGNOSIS GUIDELINES & STANDARDS

**(1) International League Against Epilepsy (ILAE):** ILAE Fisher (2017) provides the most authoritative global standards for epilepsy. Its guidelines cover definitions, classifications, diagnostic criteria, EEG standards, and therapeutic approaches. The online resources are highly comprehensive, including disease definitions, taxonomies, diagnostic workflows, EEG interpretation standards, and clinical reporting recommendations.

**(2) American Epilepsy Society (AES):** AES Krumholz et al. (2015) has issued multiple evidence-based clinical guidelines addressing epilepsy diagnosis and management strategies. These documents are designed to support clinical practice and provide practical recommendations for patient care across different healthcare settings.

**(3) United Kingdom National Institute for Health and Care Excellence (NICE):** The most recent guideline. NICE NG217: Epilepsies in children, young people, and adults (2025) Jones et al. (2023), offers systematic recommendations spanning diagnosis, evaluation, treatment, and long-term management. This guideline is distinguished by its clarity, structured organization, and emphasis on clinical applicability. NICE also provides transparent documentation on its methodology for guideline development, which enhances its reproducibility and credibility.

**(4) Scottish Intercollegiate Guidelines Network (SIGN):** SIGN Soiza & Myint (2019) guideline for Epilepsy in children and young people is widely adopted in clinical practice. It exemplifies rigorous methodology and systematic development, and serves as a benchmark for evidence-based standards in pediatric and adolescent epilepsy care. In addition, other European initiatives such as EpiCARE were also consulted to ensure broader coverage of regional practices.

**(5) Japanese Society of Neurology:** The Japanese guidelines on epilepsies provide further authoritative recommendations, with a dedicated set of documents addressing diagnostic standards and management strategies Chen et al. (2021). The full guideline is available in PDF format and complements the global standards with country-specific perspectives.

# B  DETAILED DESCRIPTION OF EXPERIMENTAL DATASETS

**TUSZ Database.**  We conduct experiments on the publicly available EEG seizure benchmark dataset: the Temple University Hospital EEG Seizure Corpus (TUSZ) v1.5.2 Shah et al. (2018). TUSZ is currently one of the largest clinical EEG corpora for seizure detection, comprising 5,612 EEG recordings and 3,050 seizure annotations. It includes 19 EEG channels recorded using the standard 10-20 system, covering a broad range of subjects across diverse clinical conditions. The seizure onset-offset labels and event type annotations provided in TUSZ dataset are also employed for interpretability analysis.

**CHB-MIT Dataset.**  We additionally evaluate EEG-RAGNet on the CHB-MIT Scalp EEG Database Shoeb (2009), a widely used benchmark for pediatric epilepsy research. The dataset contains long-term scalp EEG recordings from 23 pediatric subjects with intractable seizures, sampled at 256 Hz using the international 10-20 electrode system. Each recording includes detailed seizure onset and offset annotations verified by clinical experts. Compared to TUSZ, CHB-MIT features patient-specific variability and a smaller recording scale, providing a complementary setting to assess the generalizability of EEG-RAGNet across distinct clinical populations.

## C   DESIGNED PROMPT

---

**Named Entity Recognition for EEG Epilepsy Knowledge Base**

You are a professional biomedical NER (Named Entity Recognition) system.

Given the following medical text, extract a comprehensive list of **key biomedical entities**. These should include:

1. Diseases / Disorders – e.g., epilepsy, Lennox-Gastaut syndrome.
2. Symptoms / Clinical manifestations – e.g., seizures, aura, headache.
3. Diagnostic methods / tests – e.g., EEG, MRI, CT scan.
4. Treatments / Interventions – e.g., vagus nerve stimulation, resective surgery.
5. Medications / Drugs – e.g., valproate, lamotrigine.
6. Body parts / Anatomical structures – e.g., temporal lobe, hippocampus.
7. Risk factors – e.g., family history, prenatal injury.
8. Causes / Etiologies – e.g., brain tumor, traumatic brain injury.
9. Biomarkers – e.g., interictal discharges, spike-wave complexes.
10. Biological processes / mechanisms – e.g., neuronal hyperexcitability, GABA inhibition.
11. Clinical findings / measurements – e.g., slowed response, focal spikes.
12. Procedures / Medical actions – e.g., implantation, resection, monitoring.

Return your output as a **strict JSON object**, with keys as categories and values as arrays of extracted entity strings.

Do NOT include explanations. Do NOT add markdown. Just return pure JSON.

Example format:
{
"Diseases": ["epilepsy", "Lennox-Gastaut syndrome"],
"Symptoms": ["seizures", "loss of consciousness"],
"Diagnostic_Tests": ["EEG", "MRI"],
......
}

---

**Relation Extraction for EEG Epilepsy Knowledge Base**

You are a professional biomedical relation extraction system.

Given the following medical text, extract all clinically meaningful relationships between biomedical entities. Return each relationship as a triplet in the format: (head_entity, relation, tail_entity). A relationship must reflect medical facts such as diagnosis, treatment, symptoms, causes, affected anatomy, etc.

Valid relation types include but are not limited to:

• (treatment → treats → disease).

• (test → detects → biomarker).

• (disease → presents_with → symptom).

• (drug → alleviates → symptom).

• (disease → affects → brain_region).

• (injury → causes → epilepsy).

Please return the output as a **strict JSON array of string triplets**, like:

[ ["EEG", "detects", "interictal discharges"], ["valproate", "treats", "generalized epilepsy"], ["temporal lobe", "is focus of", "partial seizures"] ]

Do NOT return explanations. Do NOT use markdown. Return JSON array only.

# D    ADDITIONAL EXPERIMENTAL RESULTS

