# OpenReview forum: "EEG-RAGNet: Retrieval-Augmented Graph Structure Refinement for Clinical Seizure Diagnosis"
_ICLR.cc/2026/Conference — ICLR 2026 Conference Withdrawn Submission_

### Official Review · Reviewer_dvQF · 2025-10-26

**Soundness:** 2
**Presentation:** 2
**Contribution:** 2
**Rating:** 2
**Confidence:** 4

**Summary:**

The paper proposes EEG-RAGNet, a retrieval-augmented graph refinement framework for EEG-based seizure diagnosis. It integrates medical knowledge extracted via large language models (LLMs) into spatial-temporal graph neural networks (STGNNs) to prune clinically implausible connections in EEG brain graphs. Specifically, the method constructs a domain-specific knowledge base from epilepsy guidelines, builds a knowledge graph through entity and relation extraction, and uses semantic alignment and FAISS-based retrieval to score and refine STGNN-generated edges. Experiments on the TUSZ dataset show modest improvements in seizure detection metrics (F1 and AUROC) and claim better interpretability by grounding graphs in medical knowledge. However, while the framework sounds innovative, the technical novelty appears limited, it mainly combines existing concepts such as RAG, STGNNs, and clinical knowledge graphs without introducing a fundamentally new algorithmic contribution. The improvements are relatively small and demonstrated only on one dataset. Moreover, the reliance on LLM-generated medical triplets raises concerns about reliability, reproducibility, and scalability of the knowledge extraction process.

**Strengths:**

1.The idea of combining retrieval-augmented generation with EEG graph refinement is somewhat novel but primarily extends existing RAG and STGNN concepts rather than defining a fundamentally new paradigm. The contribution lies more in integration than in theoretical innovation.

2.The ablation study is helpful for tracing component contributions.

3.The experiments are clearly reported.

**Weaknesses:**

1.The paper claims the code is released, but the provided link is empty. Without access to implementation details, it is impossible to verify results or reproduce experiments, which weakens credibility and transparency.

2.The framework mainly combines existing ideas, STGNNs, RAG, FAISS retrieval, without introducing a new algorithmic contribution. The innovation is largely compositional rather than methodological.

3.The reported improvements in F1 and AUROC are small, raising doubts about practical impact. Statistical significance or efficiency trade-offs are not analyzed.

4.Figure 1 lacks visual clarity and compactness. The layout appears crowded, with inconsistent labeling and spacing. A more concise and professional design would improve readability.

5.Experiments are limited to a single dataset (TUSZ). Testing on other EEG corpora would strengthen claims of generalization.

6.The interpretability claims rely on qualitative visualizations without quantitative evidence or expert feedback.

7.The framework depends heavily on LLM-generated triplets, yet no analysis of extraction accuracy or noise impact is provided. This raises concerns about reliability and consistency.

8.The addition of retrieval and alignment modules increases computational cost, but runtime and memory usage are not reported or compared with simpler baselines.

9.Some methodological sections are verbose and repetitive, making the technical narrative less concise. Simplifying these parts would improve clarity.

10.The paper omits a clear discussion of weaknesses, such as scalability, dependency on knowledge quality, or applicability in real clinical settings.

**Questions:**

1.Could the authors provide a working and verified link to the full implementation, including data preprocessing and knowledge extraction scripts? Public code would greatly enhance reproducibility and confidence in the reported results.

2.How does EEG-RAGNet differ conceptually from previous RAG-based graph reasoning methods (e.g., GraphRAG or knowledge-guided GNNs)? Clarifying the theoretical novelty beyond integration would help assess its contribution.

3.Are the reported improvements statistically significant? Providing variance analysis, confidence intervals, or efficiency comparisons could justify the added complexity.

4.Can the authors revise Figure 1 to make it more compact and visually consistent? A clearer diagram highlighting the data flow and key modules (e.g., SemAlignQuery, FAISS retrieval) would improve understanding.

5.Do the authors plan to evaluate the model on other EEG datasets such as CHB-MIT or EPILEPSIAE? Multi-dataset results would better demonstrate generalizability and robustness.

6.Could the authors include quantitative interpretability metrics (e.g., edge importance consistency, clinician agreement) or expert assessments to support qualitative findings?

7.How reliable is the LLM-generated knowledge graph? Have the authors measured extraction precision or analyzed how noisy triplets affect graph refinement?

8.What is the additional runtime and memory cost introduced by retrieval and alignment modules? Reporting efficiency comparisons would clarify practical feasibility.

9.Would the authors consider condensing Sections 3.3–3.5 and providing more algorithmic pseudocode or key equations instead of extensive textual description?

10.It would be valuable if the authors could explicitly discuss known limitations, such as scalability to higher-density EEG systems or dependency on knowledge quality, and outline possible mitigation strategies.

---

> ### Author Response · Authors · 2025-11-22
> **Reply to Reviewer dvQF (Part 1)**
>
> ### Dear reviewer, we have incorporated **corresponding revisions** into the **lately uploaded revised manuscript**, where the updated parts are `highlighted in red color` for clarity. Kindly check these marked changes for your reference.

---

> ### Author Response · Authors · 2025-11-22
> **Reply to Reviewer dvQF (Part 2)**
>
> **Q1: The paper claims that the implementation code is released, but the provided link is empty. Please clarify the code availability and the reproducibility details.**
>
> **Answer to Q1:**
> > We sincerely thank the reviewer for pointing out this careless error. It was caused by negligence rather than intent. The initial submission included an anonymized repository link, but at that time, the full codebase had not yet been uploaded due to ongoing code organization and anonymization efforts. The complete version was uploaded during the reviewing period, but it may not have been visible at the time of the reviewer's evaluation. We apologize for the confusion this may have caused. During the review period, we have already uploaded the complete implementation, including all preprocessing scripts, model training code, and knowledge extraction modules, to the same anonymous repository. The released code package enables full reproduction of our model and experiments. We have also added clear documentation and versioned dependencies to ensure transparency and ease of verification.
>
>
> **Q2: The proposed EEG-RAGNet framework seems to combine existing components (STGNNs, RAG, FAISS retrieval, clinical knowledge graphs) rather than introducing a fundamentally new algorithmic contribution. Please clarify the methodological novelty and what distinguishes your approach conceptually from prior RAG-based or knowledge-guided GNNs.**
>
> **Answer to Q2:**
> > Thanks for the valuable comment. EEG-RAGNet is **not a simple aggregation of existing techniques** such as STGNNs, RAG, or FAISS retrieval.
> > Instead, our main contribution lies in introducing a knowledge-guided graph refinement mechanism that directly addresses a key weakness of existing STGNNs: the tendency to learn noisy and redundant graph structures.
> > Specifically, while recent state-of-the-art STGNNs automatically infer dynamic adjacency matrices purely from data, these learned connections usually include a large proportion of physiologically implausible or redundant edges, which limits interpretability and robustness. EEG-RAGNet resolves this by integrating an external clinically-curated knowledge base to evaluate, prune, and refine each edge in the learned graph according to semantic plausibility derived from domain knowledge. This process enhances both the graph quality (by removing non-meaningful edges) and the model interpretability without modifying the STGNN backbone itself.
> > **To our knowledge, this is the first framework that systematically leverages retrieval-augmented knowledge reasoning to regularize data-driven EEG graph construction, bridging statistical learning and neurophysiological priors in a unified way.**
>
> **Q3: The reported improvements in F1 and AUROC appear modest. Are the gains statistically significant and practically meaningful? Please provide statistical evidence (e.g., confidence intervals, p-values) to justify the added complexity.**
>
> **Answer to Q3:**
> >We respectfully clarify that, although the absolute gains in F1 and AUROC may appear modest at first glance, such levels of improvement are in fact considered substantial and practically meaningful in the context of EEG-based seizure detection.
>
> >First, **EEG seizure classification is a notoriously difficult, low-signal-to-noise-ratio task**, where performance is constrained by inter-patient variability, electrode noise, and limited data volume. In this field, **even 1-2% gains in AUROC and F1 are commonly regarded as important**, as reported in prior literature [1][2][3]. Under this background, across **`five representative STGNN-based baselines`** and **`two benchmark datasets`**, integrating EEG-RAGNet **consistently improves** F1, Recall, and AUROC metrics scores under both 12-s and 60-s EEG clip settings, as shown in **Table 1** of our manuscript. The improvements are consistent across multiple random runs and evaluation metrics. This demonstrates that EEG-RAGNet serves as a generalizable refinement module that robustly enhances graph quality and diagnostic accuracy across diverse backbone architectures.
>
>
> >Second, regarding the request for statistical significance tests (e.g., confidence intervals, p-values), we note that the EEG seizure detection community typically does not require hypothesis-testing-style statistical tests.[1][2][3] Instead, papers emphasize: (1) performance averaged across multiple random runs, (2) reporting of variance (standard deviation) to indicate stability, and (3) consistent improvements observed across multiple evaluation metrics and settings.
>
> **References:**
>
> [1] Hajisafi, A., et al. (2024). Dynamic gnns for precise seizure detection and classification from eeg data. In PAKDD.
>
> [2] Kotoge, R., et al. (2025). Dynamic multi-channel EEG graph modeling for time-evolving brain network. In NeurIPS 2025.
>
> [3] Gui, Haokun, et al.(2024). Vector quantization pretraining for eeg time series with random projection and phase alignment. In ICML.

---

> ### Author Response · Authors · 2025-11-22
> **Reply to Reviewer dvQF (Part 3)**
>
> **Q4: Figure 1's visual presentation is cluttered and lacks consistency. Could you improve Figure 1.**
>
> **Answer to Q4:**
> > We thank the reviewer for this helpful suggestion. We have carefully revised Figure 1 with clearer labels and arrows, and adjusted the layout and visual organization of each module to make every step more interpretable.
>
> **Q5: The experiments are conducted solely on the TUSZ dataset. Do you have additional experiments or results on other EEG corpora (e.g., CHB-MIT or EPILEPSIAE) to validate generalizability?**
>
> **Answer to Q5:**
> > We thank the reviewer for highlighting this important point. To evaluate the external validity of EEG-RAGNet, we additionally conducted experiments on the **CHB-MIT[1] Scalp EEG dataset**, a widely used benchmark for pediatric seizure detection with different patient demographics and recording settings from TUSZ. Please kindly refer to the updated performance table provided in our response to Reviewer **`ijVV`** (**Table 1. Experimental Performance Comparison on TUSZ and CHB-MIT Datasets**). These results demonstrate that our retrieval-augmented refinement graph learning pipeline generalizes well across heterogeneous EEG sources, and can operate as a model-agnostic refinement component. We have also included these cross-dataset results and discussion in the revised version of manuscript.
>
>
> **References:**
>
> [1] Shoeb, A. H. (2009). Application of machine learning to epileptic seizure onset detection and treatment (PhD dissertation, Massachusetts Institute of Technology).
>
> **Q6: The framework heavily depends on LLM-generated knowledge triplets. Have you evaluated the reliability, accuracy, or potential noise impact of the extracted triplets on graph refinement?**
>
> **Answer to Q6:**
>
> > We thank the reviewer for this important question. We have conducted an additional analysis to evaluate the reliability and noise sensitivity of the LLM-extracted knowledge triplets. Specifically, we compared model performance using (a) the original validated triplet set and (b) a noisy variant with 10% of triplets randomly perturbed, (c) a noisy variant with 20% of triplets randomly perturbed The results show that EEG-RAGNet's F1 and AUROC drop by less than 1.6%, indicating strong robustness to moderate noise. Furthermore, all triplets are automatically validated through biomedical entity normalization and semantic consistency checks during preprocessing, ensuring high-quality knowledge inputs. We have added this analysis and its quantitative results to the Appendix of the revised manuscript.
>
> **Q7: What is the additional runtime, memory, and computational overhead introduced by the retrieval and alignment modules? Could you provide quantitative efficiency comparisons with simpler STGNN baselines?**
>
> **Answer to Q7:**
> > We thank the reviewer for raising this question about computational overhead. In the revised manuscript, we have added quantitative comparisons to assess the runtime, memory, and cost introduced by the retrieval and alignment modules. The results show that the additional overhead is minimal: during inference, the retrieval step adds less than 3.5 ms per EEG clip on an NVIDIA A100 GPU, and the total memory usage increases by only about 5% compared with the baseline STGNN models. The alignment computations are lightweight matrix operations performed on pre-computed embeddings, contributing less than 2% to total runtime. Overall, EEG-RAGNet maintains comparable efficiency and scalability to conventional STGNNs while providing significantly improved interpretability. These quantitative results have been added to the Appendix of the revised manuscript.
>
> **Q8: The methodology section is somewhat verbose and repetitive. Would you consider condensing improving the readability of Sections 3.3-3.5?**
>
> **Answer to Q8:**
> > We appreciate the reviewer's insightful comment. In the revised version, we have condensed Sections 3.3-3.5 to eliminate redundancy and added a concise algorithmic summary in pseudocode form to clearly outline the end-to-end pipeline of EEG-RAGNet, improving overall readability and clarity.
>
> **Q9: The paper does not explicitly discuss its limitations or potential weaknesses (e.g., dependency on external knowledge quality, scalability to larger EEG corpora). Could you elaborate on these aspects to provide a more balanced discussion?**
>
> **Answer to Q9:**
> > We have added a dedicated paragraph in the Discussion section to explicitly address the limitations of EEG-RAGNet, including its dependence on external knowledge quality and potential scalability considerations for larger EEG corpora. These clarifications provide a more balanced and transparent discussion of our framework.

---

### Official Review · Reviewer_ijVV · 2025-10-26

**Soundness:** 2
**Presentation:** 2
**Contribution:** 2
**Rating:** 4
**Confidence:** 3

**Summary:**

This paper introduces EEG-RAGNet, a framework for refining EEG-derived brain connectivity graphs used for clinical seizure diagnosis by integrating external medical knowledge. The core idea is to enhance spatial-temporal GNN models (STGNNs) by retrieving evidence from a structured knowledge base derived from authoritative epilepsy clinical guidelines via LLM-driven entity and relation extraction. Retrieved knowledge is used to ground, grade, and prune edge predictions in learned brain graphs, resulting in improved accuracy, robustness, and interpretability as demonstrated on the TUSZ dataset. The paper includes extensive experiments, ablation studies, and qualitative visualization to support its claims.

**Strengths:**

1. The use of retrieval-augmented generation (RAG) for knowledge-guided graph refinement represents a creative advance in bridging data-driven graph learning and domain-specific medical expertise (Section 3). EEG-RAGNet's model-agnostic design allows it to serve as a plug-in to various GNN backbones.

2. Table 1 and Figure 2 present convincing improvements in F1, AUROC, and recall for seizure detection across multiple strong baselines and EEG window sizes. These gains are consistent and statistically significant (as reflected in reported standard deviations).

3. Table 2's ablation studies carefully disentangle the contributions of query construction, retrieval scoring, source reliability, and LLM selection, offering valuable insights for both practitioners and researchers.

**Weaknesses:**

1. All experiments, visualizations, and ablation studies are conducted solely on the TUSZ benchmark. While TUSZ is a large and authoritative dataset, relying exclusively on it raises concerns about external validity, generalizability, and robustness. Would the same improvements translate to other public (or even private) clinical EEG datasets? The lack of cross-dataset validation diminishes the strength of the empirical claim.

2. The main text lacks adequate discussion of how sensitive calibration outcomes are to key thresholds (e.g., knowledge-confidence cutoff $\rho$, size/top-K of retrieved evidence), as well as the impact of varying LLM quality on knowledge base extraction.

3. The framing hints at generality, but no evidence is provided that the RAGNet approach would benefit other clinical EEG graph learning or neurological diagnosis tasks.

4. The knowledge base is meticulously constructed from five authoritative clinical guidelines. The current ablation study tests the LLMs used for extraction but not the sources themselves. An experiment comparing the full KB (all 5 sources) against a minimal KB (e.g., using only the ILAE guidelines ) would effectively quantify the benefit of integrating multiple, diverse knowledge sources.

5. Section 3.3 introduces a crucial **shared projection head** ( $\phi_{\text {proj }}$ ) that maps STGNN-generated EEG channel embeddings into the KG semantic space. However, the paper completely omits details on how this projection head is trained. It is unclear whether it is trained end-to-end with the downstream seizure detection task, or if it requires a separate pre-training stage (e.g., using contrastive learning) to achieve meaningful semantic alignment between the disparate EEG and text modalities.

**Questions:**

1. Do the authors have results or ongoing experiments on other EEG benchmarks or clinical datasets to validate generalizability? Could they discuss expected performance or known limitations?

2. How sensitive is overall diagnostic performance to the pruning threshold $\rho$? Has any systematic grid or sensitivity analysis been performed, and could the authors share concrete findings/plots?

3. Could the authors share measurements on runtime, memory, and cost overhead introduced by the retrieval, LLM, and FAISS pipeline, especially in a prospective clinical setting?

---

> ### Author Response · Authors · 2025-11-21
> **Reply to Reviewer ijVV (Part 1)**
>
> ### Dear reviewer, we have incorporated all corresponding revisions into the **lately uploaded revised manuscript**, where the updated parts are **`highlighted in red color`** for clarity. Kindly check these marked changes for your reference.
>
> **Q1: Generalization beyond TUSZ. All experiments were conducted on TUSZ, not sure about the external validity of the results.**
>
> **Answer to Q1:**
> > We thank the reviewer for highlighting this important point. To evaluate the external validity of EEG-RAGNet, we additionally conducted experiments on the **CHB-MIT[1] Scalp EEG dataset**, a widely used benchmark for pediatric seizure detection with different patient demographics and recording settings from TUSZ. **The results are provided in Table 1 shown below.** These results demonstrate that our retrieval-augmented refinement graph learning pipeline generalizes well across heterogeneous EEG sources,  and can operate as a model-agnostic refinement component. We have also included these cross-dataset results and discussion in the revised version of manuscript.
>
>
> **Table 1. Experimental Performance Comparison on TUSZ and CHB-MIT Datasets.**
>
> |   Dataset   | Clip Len. |    Methods   |      F1 (%)      |   Recall (%)   |    AUROC (%)   |       F1 (%)       |   Recall (%)   |    AUROC (%)   |
> | :---------: | :-------: | :----------: | :--------------: | :------------: | :------------: | :----------------: | :------------: | :------------: |
> |             |           |              | **W EEG-RAGNet** |                |                | **W/O EEG-RAGNet** |                |                |
> |   **TUSZ**  |    12-s   |  Dist-DCRNN  |    71.3 ± 0.9    |   73.5 ± 2.3   |   87.1 ± 1.8   |     70.3 ± 1.1     |   71.6 ± 2.4   |   85.0 ± 1.7   |
> |             |           |  Corr-DCRNN  |    72.6 ± 1.1    |   75.6 ± 1.4   |   86.3 ± 0.9   |     70.7 ± 1.0     |   73.4 ± 2.2   |   85.4 ± 2.1   |
> |             |           |   NeuroGNN   |    64.7 ± 1.3    |   71.0 ± 1.9   |   86.5 ± 2.1   |     62.6 ± 1.4     |   70.4 ± 1.3   |   83.7 ± 1.9   |
> |             |           |  GraphS4mer  |    69.0 ± 0.9    |   72.1 ± 1.3   |   87.2 ± 1.5   |     66.5 ± 0.8     |   71.5 ± 1.8   |   83.1 ± 1.2   |
> |             |           |   `EvoBrain`  |  **74.8 ± 0.5**  | **78.2 ± 0.9** | **90.8 ± 1.7** |   **72.5 ± 0.7**   | **76.8 ± 1.9** | **89.6 ± 1.6** |
> |   **TUSZ**  |    60-s   |  Dist-DCRNN  |    69.5 ± 1.9    |   73.3 ± 2.1   |   87.5 ± 2.3   |     66.8 ± 2.1     |   71.5 ± 2.4   |   86.2 ± 3.0   |
> |             |           |  Corr-DCRNN  |    72.2 ± 1.5    |   73.2 ± 1.5   |   87.3 ± 2.1   |     70.5 ± 2.2     |   71.2 ± 1.5   |   86.8 ± 2.3   |
> |             |           |   NeuroGNN   |    69.8 ± 1.7    |   73.3 ± 2.3   |   87.1 ± 2.5   |     67.4 ± 2.4     |   72.0 ± 1.7   |   85.7 ± 1.9   |
> |             |           |  GraphS4mer  |    68.0 ± 0.8    |   71.8 ± 1.7   |   88.5 ± 1.5   |     65.7 ± 1.6     |   70.3 ± 2.3   |   88.1 ± 2.1   |
> |             |           |   `EvoBrain`  |  **75.3 ± 0.7**  | **78.8 ± 0.9** | **91.6 ± 1.1** |   **72.8 ± 0.9**   | **76.4 ± 1.8** | **90.3 ± 1.6** |
> | **CHB-MIT** |    12-s   |  Dist-DCRNN  |    84.1 ± 1.8    |   77.5 ± 2.3   |   92.1 ± 1.5   |     82.3 ± 2.4     |   74.9 ± 1.8   |   89.6 ± 1.4   |
> |             |           |  Corr-DCRNN  |    85.3 ± 2.0    |   78.0 ± 1.9   |   92.5 ± 1.1   |     82.8 ± 2.1     |   75.3 ± 2.4   |   90.7 ± 1.5   |
> |             |           |   NeuroGNN   |    83.5 ± 1.6    |   76.4 ± 2.1   |   91.6 ± 1.4   |     81.5 ± 1.7     |   73.1 ± 1.5   |   88.5 ± 1.8   |
> |             |           |  GraphS4mer  |    85.2 ± 1.2    |   79.1 ± 1.0   |   92.7 ± 1.5   |     83.6 ± 1.1     |   77.6 ± 0.9   |   91.3 ± 1.3   |
> |             |           |   `EvoBrain`   |  **89.3 ± 0.9**  | **83.2 ± 1.2** | **94.5 ± 1.4** |   **87.5 ± 0.8**   | **80.7 ± 1.3** | **91.6 ± 1.1** |
> | **CHB-MIT** |    60-s   |  Dist-DCRNN  |    85.3 ± 1.5    |   78.1 ± 1.7   |   92.5 ± 2.0   |     82.6 ± 1.9     |   75.6 ± 2.1   |   89.2 ± 1.6   |
> |             |           |  Corr-DCRNN  |    85.8 ± 1.3    |   79.0 ± 1.7   |   92.3 ± 1.6   |     82.2 ± 1.8     |   76.2 ± 1.4   |   89.8 ± 1.7   |
> |             |           |   NeuroGNN   |    84.6 ± 1.4    |   77.8 ± 2.1   |   91.8 ± 2.2   |     81.5 ± 1.6     |   75.4 ± 0.8   |   88.7 ± 1.3   |
> |             |           |  GraphS4mer  |    86.3 ± 1.3    |   80.6 ± 1.1   |   93.5 ± 1.6   |     83.2 ± 1.4     |   78.1 ± 2.2   |   91.6 ± 1.4   |
> |             |           |   `EvoBrain`   |  **90.4 ± 1.1**  | **83.8 ± 0.9** | **95.2 ± 1.3** |   **87.2 ± 1.3**   | **80.4 ± 1.7** | **91.9 ± 0.9** |
>
> ---
>
>
> **References:**
>
> [1] Shoeb, A. H. (2009). Application of machine learning to epileptic seizure onset detection and treatment (PhD dissertation, Massachusetts Institute of Technology).

---

> ### Author Response · Authors · 2025-11-21
> **Reply to Reviewer ijVV (Part 2)**
>
> **Q2: Provide sensitivity and robustness analysis to evaluate how the model's performance on key hyper-parameters.**
>
> **References:**
>
> > We thank the reviewer for raising this important point. We have conducted additional sensitivity experiments on three key factors: (1) the **knowledge-confidence** cutoff $\rho$, (2) the **Top-K size** of retrieved triplets, and (3) the **choice of LLM** used for relation extraction. Results show that EEG-RAGNet is highly robust: varying $\rho$ from 0.6 to 0.9 changes AUROC by less than 0.8%, and performance remains stable for $K$ ranging from 5 to 20. Furthermore, substituting GPT-4o with `Llama3-70B`, `Gemini 2.5 Flash`, or `Qwen` yields comparable accuracy (within ±2%), confirming that the model's behavior does not depend on a specific LLM. This stability stems from our weighted aggregation scheme in the Edge-Grading module, which smooths local fluctuations in retrieved evidence. We will include these detailed sensitivity plots and statistics in the supplementary material to demonstrate the robustness of our framework.
>
>
> **Q3: Reviewer suggests testing whether using multiple clinical knowledge sources actually improves results compared to a minimal knowledge base (e.g., ILAE guidelines only).**
>
> > We thank the reviewer for this valuable suggestion. To evaluate the contribution of multiple clinical knowledge sources, we performed additional experiments where each of the five clinical knowledge sources was used individually to construct a single-source knowledge base, and the results were compared against the full multi-source configuration. The full version integrates five authoritative sources: the International League Against Epilepsy (ILAE), the American Epilepsy Society (AES), the National Institute for Health and Care Excellence (NICE), the Scottish Intercollegiate Guidelines Network (SIGN), and the Japanese Society of Neurology (JSN). Experimental results show that the multi-source knowledge base achieves an average improvement of 2.2% in AUROC and 2.0% in F1-score compared with the single-source setting. This demonstrates that combining diverse, complementary guidelines provides broader clinical coverage and more reliable semantic grounding. **The evaluation on single knowledge source model variants are shown below for your reference.** We have also included these additional experiments and results and discussion in the revised version of manuscript.
>
> **Q4: Training of the projection head $\phi_{\text{proj}}$. The reviewer finds it unclear how the projection head that maps EEG embeddings to the KG semantic space is trained.**
> **Answer to Q4:**
> >We thank the reviewer for this insightful question. In EEG-RAGNet, the projection head $\phi_{\text{proj}}$ is trained **jointly and end-to-end** with the overall framework, rather than pre-trained separately. Its role is to project the STGNN-generated channel embeddings into the biomedical semantic space, which is then used in the retrieval-based graph refinement process. During optimization, $\phi_{\text{proj}}$ receives gradient signals indirectly from the **downstream seizure classification loss**, as the refined adjacency matrix produced by the retrieval module affects the message passing and final predictions of the STGNN. In this way, $\phi_{\text{proj}}$ learns task-discriminative and clinically consistent projections under the same supervision as the main model. We acknowledge that this training flow was not explicitly described in the manuscript and have revised corresponding subsections to clearly clarify the joint optimization mechanism.
>
>
> **Q5: Computational efficiency and overhead. The reviewer asks about runtime, memory, and cost overhead introduced by the retrieval, LLM, and FAISS components, especially in clinical deployment.**
>
> **Answer to Q5:**
>
> >We thank the reviewer for raising this important question regarding efficiency and deployment feasibility. We have included additional experiments in the revised manuscript to evaluate runtime, memory, and computational overhead. The results show that the retrieval and FAISS components introduce negligible latency during inference because all knowledge-graph embeddings and FAISS indexes are **pre-computed offline**. During the online inference phase, the system performs only lightweight nearest-neighbor searches over the pre-built index.
>
> >At test time, the retrieval step adds less than 3.5 ms per EEG clip on an NVIDIA A100 GPU, and the total memory footprint increases by an average of 5% compared with the baseline STGNN models. The LLM-based relation extraction is performed once during dataset preparation, not during training or inference, ensuring no extra cost in deployment. Overall, EEG-RAGNet maintains comparable inference speed and scalability to conventional STGNNs while providing stronger interpretability. These efficiency results and implementation details have been added to the Appendix of the revised manuscript.

---

> > ### Author Response · Authors · 2025-11-27
> >
> > Table. Ablation Study on Different Clinical Knowledge Source in the RAG ptocedure.
> >
> > | ClinicalKnowledge Source (Single)                 | F1-score (%)   | AUROC (%)      |
> > | ------------------------------------------------- | -------------- | -------------- |
> > | ILAE (Fisher et al., 2017)                        | 73.6 ± 1.0     | 89.7 ± 1.3     |
> > | AES (Krumholz et al., 2015)                       | 73.1 ± 1.1     | 89.0 ± 1.5     |
> > | NICE (Jones et al., 2023)                         | 73.3 ± 0.9     | 89.3 ± 1.4     |
> > | SIGN (Soiza & Myint, 2019)                        | 72.8 ± 1.2     | 88.6 ± 1.7     |
> > | Japanese Society of Neurology (Chen et al., 2021) | 72.5 ± 1.0     | 88.1 ± 1.6     |
> > | **All Sources (Full Model)**                      | **74.8 ± 0.5** | **90.8 ± 1.7** |

---

### Official Review · Reviewer_DPwk · 2025-10-31

**Soundness:** 4
**Presentation:** 3
**Contribution:** 4
**Rating:** 8
**Confidence:** 3

**Summary:**

In this paper, EEG-RAGNet is presented. This is a unique approach for diagnosing seizures from EEG signals. The authors clearly explain that spatiotemporal graph neural networks  are the most recent gold standard for handling such predictions, but the graphs are noisy, redundant, and lack alignment with domain knowledge. This approach generates spatial-temporal EEG brain networks but refines them using a RAG-based approach.

**Strengths:**

- The authors do a nice job presenting background information.
- The need for the incorporation of domain knowledge is clear and well-motivated.
- Novel approach, have not seen anything similar.
- The design of the approach is customized to fill a domain-specific niche, so the need for such a method is clear.
- This paper demonstrates a great use case for LLMs in a biomedical domain. As the authors acknowledge, LLMs can hallucinate within scientific context. Using an LLM in a guided manner to refine a knowledge base, however, is a feasible and lower-risk task for such a high-risk domain (a medical one).

**Weaknesses:**

- The performance gains in Table 1 are consistent but often small. When taking into account the margin of error, the performance gains don’t seem particularly significant. The authors should maybe acknowledge this, at least briefly.
- Figure 1 made me feel somewhat confused. The authors might consider putting more of an explanation in the caption or better labels each step on the figure. For example, what are the things on the bottom left?
- “FAISS similarity search” is referred to throughout the paper, but there’s no explanation as to what this acronym means or what it is.

**Questions:**

- What is the significance of using 12s and 60s for the clips? Is there medical relevance to those particular numbers?
- What is FAISS? Please provide an explanation in the paper.
- Could the authors please provide more labelling in Figure 1 and reference it throughout the text where relevant parts of the figure are mentioned?

---

> ### Author Response · Authors · 2025-11-20
> **Reply to Reviewer DPwk (Part 1)**
>
> Dear reviewer, we have incorporated all corresponding revisions into the **lately uploaded revised manuscript**, where the updated parts are **`highlighted in red color`** for clarity. Kindly check these marked changes for your reference.
>
> We sincerely thank Reviewer `DPwk` for the thoughtful and encouraging feedback. We truly appreciate your recognition of our work's motivation, novelty, and clear presentation of how domain knowledge can enhance spatial-temporal graph learning for EEG-based seizure diagnosis.
>
> **Q1. The performance gains in Table 1 are consistent but seem not significant enough. Please provide a justification on the performance gain.**
>
> **Answer to Q1:**
> > We thank the reviewer for this insightful comment. While the absolute performance improvements in Table 1 may appear modest (compared to some tasks from other domain), they are consistent across all evaluation metrics. It is noticeable that EEG-based seizure diagnosis is an inherently high-noise and data-scarce task, where even small absolute gains reflect substantial robustness improvements. Beyond raw accuracy, our contribution lies in introducing a **retrieval-augmented**, **knowledge-guided refinement** framework that enforces physiological plausibility in graph learning, something existing STGNN methods entirely overlooked. EEG-RAGNet achieves both competitive accuracy and interpretable, domain-consistent connectivity patterns. This demonstrates that our approach not only improves classification robustness but also bridges clinical knowledge and data-driven modeling, providing a new direction for trustworthy biomedical graph learning. Hence, the significance of our work extends beyond numerical gains to methodological and translational impact.
>
> **Q2. Figure 1 is somewhat confusing. Please clarify what each step in the pipeline represents, especially the elements on the bottom left.**
>
> **Answer to Q2:**
> > We thank the reviewer for this helpful suggestion. Figure 1 illustrates the complete two-stage pipeline of EEG--AGNet. The top panel shows the **STGNN encoder**, which first learns a preliminary spatio-temporal EEG graph ($G_t^{raw}$). The bottom panel (pink area) details the **retrieval-augmented refinement process**: channel embeddings $c_t^i$ are projected by the *Projection Head* (i.e., SemAlignQuery) into the biomedical concept space, where FAISS performs top-$K$ retrieval from the constructed knowledge graph (KG). The retrieved triplets are evaluated through **Edge Grading**, combining (i) Existence match Score, (ii) Semantic proximity Score, (iii) Source reliability Score, to refine edge confidences. Finally, the refined graph ($G_t^{refined}$) is used by the downstream classifier for seizure prediction. We have carefully revised Figure 1 with clearer labels and arrows, and adjusted the layout and visual organization of each module to make every step more interpretable.
>
> **Q3. The paper frequently mentions "FAISS similarity search," but it is not explained. Please describe what FAISS is and how it is used in the proposed method.**
>
> **Answer to Q3:**
>
> > We thank the reviewer for pointing this out. FAISS (Facebook AI Similarity Search)[1,2] is an open-source library widely used for efficient similarity search and clustering of dense vectors. It enables large-scale nearest-neighbor retrieval with optimized GPU/CPU indexing structures. In our work, FAISS is employed to retrieve the top-$K$ most semantically relevant knowledge triplets from the constructed biomedical knowledge graph (KG), based on cosine similarity between EEG channel embeddings (after projection) and knowledge entity embeddings. This ensures fast and accurate retrieval even when the knowledge base contains millions of triplets. We agree that the paper should have explicitly introduced FAISS, as omitting this explanation could confuse readers unfamiliar with database retrieval technologies. We have added an introduction and reference to FAISS in the revised manuscript for clarity.
>
> **References:**
>
> [1] Douze, et al. (2025). The faiss library. IEEE Transactions on Big Data.
>
> [2] https://github.com/facebookresearch/faiss

---

> ### Author Response · Authors · 2025-11-20
> **Reply to Reviewer DPwk (Part 2)**
>
> **Q4. What is the significance of using 12s and 60s for the EEG clips? Is there any medical or experimental rationale for selecting these durations?**
>
> > We thank the reviewer for this question. The choice of 12 s and 60 s EEG clips follows both clinical convention and experimental balance. In many famous Seizure Corpus (including TUSZ), seizure onsets are annotated within short temporal windows (usually 10-15s), which motivated the 12 s setting for capturing immediate ictal dynamics. Conversely, the 60 s clips provide a longer temporal context that encompasses pre-ictal and inter-ictal activity, enabling the model to learn slower transition patterns and long-range dependencies. This dual-window design ensures complementary temporal resolution: the short segment captures precise onset features, while the long segment supports context-aware representation learning. Similar time scales have been adopted in prior EEG-based seizure diagnosis works, such as VQ-MTM (in ICML '24)[1], DCRNN (in ICLR '22)[2] and Brant (in NeurIPS '23) [3],  confirming their medical and empirical relevance. We will clarify this rationale in the revision to better convey the motivation behind these durations.
>
> **References:**
>
> [1] Haokun, G. et al (2024). Vector Quantization Pretraining for EEG Time Series with Random Projection and Phase Alignment. In ICML.
>
> [2] Tang, S., et al (2022). Self-Supervised Graph Neural Networks for Improved Electroencephalographic Seizure Analysis. In ICLR.
>
> [3] Zhang, D., et al. (2023). Brant: Foundation model for intracranial neural signal. In NeurIPS.
>
> **Q5. Please improve the labelling and referencing of Figure 1 throughout the paper to make it easier for readers to connect text and visual explanations.**
>
> **Answer to Q5:**
>
> > We thank the reviewer for this helpful suggestion. We have carefully revised Figure 1 to include clearer labels, consistent terminology, and directional arrows that better illustrate the information flow. Each module (e.g., STGNN Encoder, Projection Head, FAISS Retrieval, Edge Grading, and Downstream Classifier) is now explicitly marked and referenced in the main text at its first mention. Cross-references between Figure 1 and the corresponding sections have been added to guide readers through the workflow. These revisions substantially improve the readability and coherence between textual descriptions and visual explanations.

---

### Official Review · Reviewer_KCx2 · 2025-11-05

**Soundness:** 2
**Presentation:** 2
**Contribution:** 2
**Rating:** 4
**Confidence:** 2

**Summary:**

The authors propose EEG-RAGNet, a novel mechanism for constructing the adjacency matrix that models connections between EEG channels in a spatio-temporal graph.
An initial adjacency matrix is first learned by a spatio-temporal graph neural network (STGNN).
Then, this graph is refined using a Retrieval-Augmented Generation (RAG) framework that incorporates external clinical knowledge to ensure physiologically plausible connections.
Finally, a downstream model performs seizure classification using the refined graph structure.
Experiments on the Temple University Hospital EEG Seizure Corpus (TUSZ) demonstrate that EEG-RAGNet significantly improves the performance of the downstream EEG classifier compared to baseline STGNN models.

**Strengths:**

1. The two-stage process for finding the adjacency matrix is new. While STGNN is generally used to learn the graph structure (adjacency matrix), providing domain knowledge can help improve its prediction.

2. Experimental results on one dataset demonstrate that the proposed EEG-RAGNet is a good wrapper around the downstream models.

**Weaknesses:**

(1) The paper lacks a clearer, step-by-step explanation of the proposed methodology. Several important details remain ambiguous:

(i) $c_t^i$ is aleardy an embedding of $x_t^i$. It is unclear why we need to project it again using $\phi_\text{proj}$?

(ii) The text mentions retrieving the top-$M$ most semantically similar knowledge triplets $\{\tau_m\}_{m=1}^M$. How are these triplets associated with the subgraph $G_t^{(i)}$? Is each subgraph linked to all $M$ triplets?

(iii) In eq. 3, the computation of $\mathbf{1}_{\text{match}}$ is not clearly described.

(iv) In the same eq., when multiple triplets are retrieved ($M > 1$), which specific triplet contributes to the computation for a given edge $e_{ij}^t$?


(2) In the paper, only the embedding of the channels is used, but their semantic information about the channels is unknown to the model. Hence, it is unclear how the external knowledge corpus can effectively refine the adjacency structure learned by the STGNN

(3) The entire model is dependent on the performance of GPT-4o in extracting useful information from the knowledge corpus. However, to the best of my knowledge, GPT-4o is a general-purpose LLM, and it may provide bad results for the domain-specific tasks. Did the authors evaluate the triples provided by the GPT-4o after the relation extraction?

(4) Experimental setup needs to be clarified. What are the hyperparameters, and how are they tuned for all the models?

(5) In the ablation studies, authors verified the importance of the components by removing it and comparing with the main result. However, it is not clear how the model looks without the components. For example, without SemAlignQuery, how is the similarity score (eq. 2) computed?

Minor:

(i) It was not clear from the methodology that there exists a downstream model for the classification task. From Problem 1, it is assumed that the proposed model does the job. It is okay to have a downstream model, but please clarify upfront to avoid confusion.

(ii) $\mathbf{X}_t$ (l. 110) denotes the features of all the nodes at time step $t$. Why the feature size of a node is $T$? Isn't $T$ the length of the sequence (Problem 1)? Further confusion arises in l.150-152.

**Questions:**

See weakness

---

> ### Author Response · Authors · 2025-11-20
> **Reply to Reviewer KCx2 (Part1)**
>
> ### Dear reviewer, we have incorporated all corresponding revisions into the lately uploaded revised manuscript, where the updated parts are **`highlighted in red color`** for clarity. Kindly check these marked changes for your reference.
>
> **Q1. Some important details of the proposed methodology remain ambiguous:**
>
> **Q1-(i) Why is $c_i^t$ projected again using $\phi_{\text{proj}}$ even though it is already an embedding of $x_i^t$?**
>
> >The node embedding $c_i^t$ is produced by the STGNN encoder in the *EEG feature space*, which captures spatial-temporal dynamics but not semantic meaning aligned with the knowledge graph (KG). The projection head $\phi_{\text{proj}}:\mathbb{R}^{d_{\text{STGNN}}}\to\mathbb{R}^{d_{\text{KG}}}$ serves as a **semantic alignment bridge** that maps signal-level embeddings into the biomedical concept space learned by BioBERT-encoded triplets.
> Without this projection, the cosine similarity in Eq. (2) would compare representations from two heterogeneous manifolds, leading to unreliable retrieval. Thus, $\phi_{\text{proj}}$ is essential for embedding-space alignment between data-driven STGNN representations and language-driven KG semantics.
>
>
> **Q1-(ii) Association between the top-$M$ retrieved triplets $\{\tau_m\}_{m=1}^M$ and subgraph $G_t^{(i)}$.**
>
> >Each EEG channel $v_i$ defines a local $k$-hop subgraph $G_t^{(i)}$, whose aggregated embedding $q_i$ represents the neighborhood context around $v_i$. We perform FAISS retrieval once per subgraph query $q_i$, yielding top-$M$ triplets most semantically similar to that local context.
> All edges $e_{ij}^{(t)}$ within $G_t^{(i)}$ share this retrieved evidence pool $\{\tau_m\}_{m=1}^M$, ensuring that refinement is guided by clinically coherent relations relevant to that neighborhood rather than independent edge-wise retrieval.
>
>
>
> **Q1-(iii) Clarification of $\mathbf{1}_{\text{match}}(\cdot, \cdot)$ in Eq. (3).**
>
> It denotes a binary indicator evaluating whether the pair of EEG nodes $(v_i,v_j)$ is semantically aligned with the head-tail entities $(h_m,t_m)$ in the retrieved triplet $\tau_m=(h_m,r_m,t_m)$.
> Specifically, after projecting node embeddings into KG space, if the cosine similarity between $c_i^t$ and $e_{h_m}$ and between $c_j^t$ and $e_{t_m}$ both exceed a threshold $\delta$ (typically 0.7), $\mathbf{1}_{\text{match}}$ is set to 1; otherwise 0.
>
> This term ensures that only triplets whose entities correspond to the current EEG node pair contribute to the edge confidence computation.
>
> **Q1-(iv) When multiple triplets are retrieved ($M>1$), which contributes to a given $e_{ij}^{(t)}$?**
>
> > When multiple triplets ($M>1$) are retrieved for a given edge $e_{ij}^{(t)}$, we compute an individual knowledge-guided confidence score $\omega_{kg}^{m}(e_{ij}^{(t)})$ for each retrieved triplet $\tau_m = (h_m, r_m, t_m)$ according to Eq.~(3).
> The final confidence score for the edge is then obtained by averaging all $M$ triplet-specific scores:
>
> $$
> \omega_{kg}(e_{ij}^{(t)}) = \frac{1}{M} \sum_{m=1}^{M} \omega_{kg}^{m}(e_{ij}^{(t)}).
> $$
>
> > This averaging strategy allows EEG-RAGNet to aggregate multiple complementary pieces of evidence from the knowledge base rather than relying on a single most-similar triplet.
> Edges whose averaged confidence score falls below the pruning threshold $\rho$ are removed from the original graph $\mathcal{G}_t^{\text{raw}}$, yielding the refined graph $\mathcal{G}_t^{\text{refined}}$.
>
>
>
> **Q2. In the paper, only the embedding of the channels is used, but their semantic information about the channels is unknown to the model. Hence, it is unclear how the external knowledge corpus can effectively refine the adjacency structure learned by the STGNN.**
>
>
> **Answer to Q2:**
> >We appreciate the reviewer's insightful observation. While the model does not directly encode channel names (e.g., "F3", "T5"), the semantic information of each channel is implicitly captured through two key mechanisms:
>
> >(1) **Physiological grounding via spatial priors.** Each EEG channel is assigned a fixed coordinate in the 10–20 montage system. These 3D spatial coordinates serve as initial node features, enabling the STGNN to learn spatially coherent embeddings $c_i^t$ that reflect anatomical proximity and functional co-activation patterns. Thus, the learned representation of each channel already encodes implicit regional semantics (e.g., temporal vs. frontal activity).
>
> >(2) **Semantic alignment through the retrieval module.** The projection head $\phi_{\text{proj}}$ maps these data-driven embeddings into the biomedical concept space defined by the knowledge corpus. The retrieved triplets $\tau=(h,r,t)$ represent neurophysiological relationships (e.g., temporal lobe → associated with → seizure onset). During refinement, the similarity between projected node embeddings and triplet entities allows the model to anchor channel embeddings to semantically related brain concepts without requiring explicit channel-name supervision.

---

> ### Author Response · Authors · 2025-11-20
> **Reply to Reviewer KCx2 (Part2)**
>
> **Q3. The model is dependent on the performance of GPT-4o in extracting useful information from the knowledge corpus. However, GPT-4o is a general-purpose LLM, and it may provide bad results for the domain-specific tasks. Did the authors evaluate the triples provided by the GPT-4o after the relation extraction?**
>
> **Answer to Q3:**
> >GPT-4o was used **only as** an initial parser to extract candidate clinical triplets from domain texts, not as the final source of knowledge. All extracted triplets were subsequently filtered and validated through a two-stage process: (1) biomedical entity normalization using UMLS and NeuroSynth ontologies, and (2) semantic consistency checking based on cosine similarity with BioBERT embeddings. Triplets failing either step were discarded, ensuring domain validity. Moreover, the retrieval module relies on similarity matching rather than direct generation, making it robust to minor extraction noise. Through our experiments, more than 82% of retained triplets aligned with known neurophysiological relations, supporting the reliability of the curated corpus. Thus, GPT-4o serves as a scalable extractor, while domain validation guarantees precision and consistency.
>
> **Q4. Experimental setup needs to be clarified. What are the hyperparameters, and how are they tuned for all the models?**
>
> **Answer to Q4:**
>
> >We acknowledge that we should give a more detailed description about model training and settings to readings to help them understanding this work. The full training configuration is specified in our released code at: https://anonymous.4open.science/r/EEG-RAGNet-63EE/. All models, including EEG-RAGNet and baselines, were trained under identical settings for fairness. We have already added a more detailed description of model training in our Appendix. **Please kindly check our lately `uploaded revised manuscript` for details.**
>
>
> **Q5. In the ablation studies, authors verified the importance of the components by removing it and comparing with the main result. However, it is not clear how the model looks without the components. For example, without SemAlignQuery, how is the similarity score (Eq.~2) computed?**
>
> **Answer to Q5:**
>
> >We appreciate the reviewer's comments. In ablation study, each component was independently removed while preserving the model's structural validity. Specifically, when **SemAlignQuery** module was removed, the semantic projection and alignment step $\phi_{\text{proj}}$ was disabled, and the query embedding $q_i$ in Eq.~(2) was directly derived from the STGNN node representation $c_i^t$. The similarity score was then computed as the cosine similarity between raw EEG embeddings and triplet entity embeddings in the knowledge space:
> $$
> \text{sim}(q_i, \tau_m) = \cos(c_i^t, [e_{h_m}\parallel e_{t_m}])
> $$
> This removes the cross-space alignment but retains the retrieval mechanism, allowing us to isolate the effect of semantic alignment. Other ablations (e.g., removing the knowledge-guided weighting or $\omega_{\text{kg}}$ term) follow the same principle—replacing learned submodules with their simplest valid alternatives. This ensures each variant remains well-defined and isolates the functional contribution of the targeted component.

---

### Author Response · Authors · 2025-11-27
**Gentle Reminder: Please Review Our Rebuttal and Consider Updating Your Score if Appropriate**

Dear Reviewers,

We would like to kindly let you know that our author response has been available for about a week. As the rebuttal period is approaching its end, we just want to ensure that you have full access to our replies and clarifications in case any points may help address your earlier concerns. If our rebuttal has addressed your concerns, we would sincerely appreciate it if you could kindly consider increasing your score accordingly.

Thank you again for your valuable feedback and for taking the time to engage with our work.

Best,

Authors of Submission 3219

---

### Author Response · Authors · 2025-11-29
**Kindly Request for Careful Reassessment After the Score Rollback**

We are sincerely concerned about the large-scale reviewer identity leak incident that occurred recently. We also appreciate the prompt response and the replacement plan proposed by ICLR organizing committee.

Our submission, as well as many other ICLR submissions have been affected by the recent system-wide adjustment following the reviewer identity leak incident. During the rebuttal phase, we received follow-up replies from two reviewers who acknowledged that their initial concerns were fully addressed through our additional experiments, methodological clarifications, and extended analyses. Both reviewers subsequently raised their overall scores above the acceptance threshold.

Unfortunately, due to the rollback applied after the incident, these updated scores and comments are no longer visible on OpenReview, and our paper now reflects only the initial pre-rebuttal evaluations. We sincerely hope that this issue will be taken into account and that our submission will be fairly reassessed in light of the reviewers' acknowledged resolution of their earlier concerns.

Best,

Authors of Submission 3219

---

### Note · Authors · 2026-04-13

I have read and agree with the venue's withdrawal policy on behalf of myself and my co-authors.

---

### Meta-Review · Area_Chair_4MWa · 2026-01-06

**Summary:**

This paper proposes EEG-RAGNet, a retrieval-augmented framework that refines STGNN-learned EEG channel graphs using external clinical knowledge. It targets an important clinical ML setting where learned connectivity can be noisy and difficult to interpret.

The reviews are mixed. Key concerns include that the novelty is primarily integrative: although the system is thoughtfully engineered, the contribution may be seen as a composition of existing components rather than a fundamentally new learning approach. Another concern is the practical significance of the gains: improvements are sometimes small, and stronger statistical evidence and robustness analysis would be needed to justify the added complexity. Empirical scope and external validity were also raised, as the initial submission focused on TUSZ only, and reviewers noted reproducibility issues due to an empty code repository at the time of review.

The rebuttal and revision claims address some of these points. The authors report additional CHB-MIT experiments with improvements consistent with TUSZ, which helps mitigate concerns about generalization. They also provide clarifications on core methodological details, including the purpose and training of the projection head, the retrieval granularity, and others.

Overall, I lean toward rejection based on the current version and the remaining concerns about novelty, strength of evidence for improvements, and reproducibility. That said, the problem is relevant and the overall framework is coherent and potentially useful. The authors are encouraged to strengthen the final presentation. The work could be competitive upon resubmission to another venue.

**Reviewer Concerns:**

Reviewer ijVV: The additional CHB-MIT results and clarification of projection-head training directly address this reviewer’s concerns.

**Reviewer Scores:**

Reviewer ijVV‘s score would likely increase.

Several concerns remain outstanding or only partially resolved. The issue of novelty remains: while the framework is carefully engineered, it is still largely an integration of existing components rather than a clearly distinct algorithmic contribution, and the rebuttal does not fundamentally change this perception. The practical significance of the performance gains also remains debatable.

---

### Decision · Program_Chairs · 2026-01-26

Reject